# Socioeconomic Enablers for Contagion: Factors Impelling the Antimicrobial Resistance Epidemic

**DOI:** 10.3390/antibiotics8030086

**Published:** 2019-06-30

**Authors:** Peter Collignon, John J. Beggs

**Affiliations:** 1Infectious Diseases Physician and Microbiologist, Australian Capital Territory Pathology, Canberra Hospital, Australian Capital Territory 2606, Australia; 2Medical School, Australian National University, Australian Capital Territory 0200, Australia; 3Monarch Institute, 10 Queen St, Melbourne 3000, Australia

**Keywords:** antimicrobial resistance, governance, water, sewage, social factors, corruption, infrastructure, antibiotics

## Abstract

Antimicrobial resistance is a growing global problem that causes increased deaths as well as increased suffering for people. Overall, there are two main factors that drive antimicrobial resistance: the volumes of antimicrobials used and the spread of resistant micro-organisms along with the genes encoding for resistance. Importantly, a growing body of evidence points to contagion (i.e., spread) being the major, but frequently under-appreciated and neglected, factor driving the increased prevalence of antimicrobial resistance. When we aggregate countries into regional groupings, it shows a pattern where there is an inverse aggregate relationship between AMR and usage. Poor infrastructure and corruption levels, however, are highly and positively correlated with antimicrobial resistance levels. Contagion, antibiotic volumes, governance, and the way antibiotics are used are profoundly affected by a host of social and economic factors. Only after we identify and adequately address these factors can antimicrobial resistance be better controlled.

## 1. Why is Antimicrobial Resistance a Major Global Concern?

Antimicrobial resistance (AMR) is a major international problem [1,2,3,4,5,6,7,8,9,10,11,12,13,14,15]. It is causing ever-increasing deaths and suffering. It is not a problem just in healthcare. In the community it is also compromising therapy, in some cases, even making it impossible to successfully treat many common bacterial infections. This includes therapy of the two bacterial pathogens causing most of the life-threatening infections that occur in people, i.e., *Staphylococcus aureus* and *E. coli*. In India with *E. coli*, because of very high antibiotic resistance rates, it appears that that about half of all community-onset urinary tract infections are, for practical purposes, untreatable [11]. In Europe people with resistant *E. coli* bloodstream infections have a mortality rate about twice that of people infected with more sensitive strains (32% vs. 17%). For those people that survive but are infected with resistant strains, they spend on average an extra five days in hospital [9].

Recently we showed that many social and economic factors are of more importance than the antibiotic volumes used in explaining why there are such marked variations in the levels of documented antibiotics resistance levels between different countries [15]. We believe these social and economic factors are often underappreciated, and even ignored by many, as contributing to the major and ever-growing global AMR problem. This invited commentary is an attempt to better highlight the importance of many of these important social and economic determinants [15] and to show why they are likely much more important to better control, than focusing on the volumes of antibiotics used, if we want to better manage and control this problem and ultimately decrease AMR levels globally. 

## 2. Causes of Antimicrobial Resistance 

Increasing AMR rates happens after bacteria change and become resistant to the antibiotics previously used to successfully treat the infections they cause. Among factors listed as causing rising prevalence of AMR, the World Health Organisation (WHO) include: over-prescribing of antibiotics, over-use of antibiotics in livestock and fish farming, poor infection control in hospitals and clinics, and lack of hygiene and poor sanitation [1]. 

Overall, as has been noted elsewhere [7], there appear to be two main factors that drive the prevalence of AMR. Both are heavily affected by social and economic determinants, and both can be much better controlled. These factors are:
(i)The volumes of antimicrobials used; and(ii)Contagion, or the spread of resistant micro-organisms and/or the genes encoding for resistance.

The development of the resistant bacteria themselves occurs by mutations or, more often, by the acquisition of resistance genes already present in other bacteria, often occurs after the uptake of mobile genetic elements such as plasmids. Spread occurs by direct human-to-human contact or via multiple vectors, such as air (e.g., respiratory droplets), water, direct animal contact, foods, insects, birds, agriculture and/or via the wider environment (especially waterways) [3,4,5,6,7,13,14,15]. This spread of already-resistant bacteria we define as “contagion”. 

A recent study we undertook, looked at what factors globally appear most to affect the resistance rates seen in different countries [15]. That study found an inverse bivariate correlation between AMR prevalence and several common economic development indicators including infrastructure, education, per capita GDP and public health spending. AMR rates were found positively correlated with higher temperature climates, poorer administrative governance, and the ratio of private to public health expenditure. When more complex analysis was done by adjusting for confounding factors using multivariable regression analysis, then better infrastructure (e.g., improved sanitation and potable water) as well as better administrative governance (e.g., less corruption) were strongly and statistically significantly associated with lower AMR indices. Surprising, and contrary to most current beliefs, antibiotic consumption was not strongly associated with AMR levels [15] either at the level of a simple bivariate correlation or in the multivariate regression analysis. This empirical evidence implies that contagion, rather than antibiotic usage volumes, is the major factor contributing to the variations in antibiotic resistant levels across countries. Contagion is, in turn, affected by many social determinants and economic factors. 

It is well known that the contagious spread of resistant bacteria (e.g., MRSA, VRE) is a major problem in nearly all health facilities. We suggest that the same phenomenon could be driving the high presence of resistance in the broad community. Poor community hygiene, in all its dimensions, enables conditions that lead to contagion within families, and then more widely. 

The contagion hypothesis predicts that higher rates of AMR will be observed in places with conditions that permit the ready spread of the resistant bugs. In such locations there will be poor access to and quality management of sanitation, water, housing, crowding, food (transport, storage, refrigeration and preparation), health facilities (hospitals, doctors, nursing, and infection control), hygiene awareness, along with weak administrative governance of many, or of all of the above. A strong contagion effect means that high prevalence of AMR can be sustained even in the presence of lower levels of antibiotic usage. Countries, regions or environments where one sees these higher rates of resistance and lower use of antibiotics are typically countries with poor infrastructure, poor water and sanitation, weak governments, and low income.

Social and economic factors are also important where they result in insufficient use of, or else poorly managed use of, available antibiotics. For example, in some instances enough antibiotics may be used to generate mutant resistant strains, but not enough or the wrong type to wipe out any resistant strains that emerge.

## 3. Resistance Levels Often Do Not Correlate with Human Usage Volumes

Aggregating countries (Figure 1) into regional groupings shows a pattern where there is an inverse aggregate relationship between AMR and usage. These data help confirm that there are other very important factors influencing AMR over and above simply antibiotic usage.

Further support for the idea that there are other, and likely more important factors, than usage volumes affecting AMR is found in Table 1. Data in the table shows higher average AMR rates in low-middle-income countries and middle-income countries (LMICs and MICs), in which per-person consumption of antibiotics is much lower than in high-income countries, [15].

Poor controls on the spread of bacteria facilitate the spread of resistant bacteria and also will facilitate the opportunity for resistance genes to insert into other bacteria. On a worldwide basis, contaminated water is likely to be the main transmission pathway. Figure 2 shows high levels of resistance in countries with poor sanitation, and these high rates of resistance persist even though these countries have low levels of antibiotic usage [15].

## 4. Antibiotic Usage Volumes are Over-Emphasized

In developed countries, most medical attention given to the rising prevalence of AMR is focused on overuse of antibiotic use by medical overprescribing as the cause. Better control of antibiotic usage, along with development of new drugs have been the priority initiatives. Most often the clarion call is “an urgent need to develop new and useful antibiotics to avoid returning to the ‘pre-antibiotic era’” [17]. 

Undeniably, selection pressure due to antibiotic exposure is an important element in the emergence and selection of AMR bacterial clones. Antibiotic consumption is an important primary factor, not only for the emergence of new antibiotic resistant strains of bacteria but, more importantly, for the selective amplification and multiplication of resistant strain of bacteria [18,19,20,21,22]. Overuse and high levels of antibiotic consumption can occur in people or in animals from overprescribing (e.g., the variations between countries in Europe) [23] or from less regulated overuse, when antibiotics are used without the need for prescription (e.g., growth promotion use in food animals, or over-the-counter and/or non-regulated antibiotic sales to people). 

In Europe, countries with higher levels of bacterial resistance exhibited significantly higher levels of per capita antibiotic use, however, the responsiveness of changes in bacterial resistance to antibiotic use was relatively low. The implication of this evidence is that other factors, and not just antibiotic usage, are playing important roles [24], with spread or “contagion” the most important (15). The European Food and Safety Authority’s extensive 2017 report concluded that the spread of drug-resistant clones impacts the relationship between antimicrobial consumption (AMC) and AMR and “could explain why significant statistical associations between AMC and AMR were sometimes found and sometimes not” [23]. 

Even in developing countries, where issues are more complex, the focus of AMR policy is usually on antibiotic volumes and healthcare facilities, rather than any focus on the broader set of social and economic determinants. Emphasis is given to the practices of health care professionals, on how patients ask for and use antibiotics, and the controls on the supply of antibiotics to the population. Areas highlighted are inapt prescription practices by doctors, lack of patient knowledge regarding use of drugs, unsanctioned sale of antimicrobials, weak and ineffective drug regulatory mechanisms, limited availability of diagnostic facilities, and non-human use of antimicrobials, such as in animal production [25]. However, these highlighted issues mainly dealing with antibiotic usage. They do not address contagion, and the associated social or economic conditions that enable the ready spread of resistant bugs. 

Antibiotic use is profoundly affected by cultural and socioeconomic circumstances [24]. While the spread of resistant bacteria affects all regions of the world, there are large disparities in country level rates of AMR [1,2,15,26]. Not all the reasons for these disparities are well understood, and many factors are responsible. Unfortunately, however, most interventions and policy approaches have been narrowly focused on the selective pressure engendered by inapt use of antibiotics. 

Lack of official government policy on the rational use of antibiotics in public and private hospitals, and irrational use in animal species may affect the prevalence of AMR [24,27]. As yet there is also no comprehensive global data on antibiotic usage in animals. 

## 5. Dissemination of Resistance: A New Focus Indicates the Major Role of Contagion

Although the use and overuse of antibiotics are primary drivers of the emergence and maintenance of AMR, other factors are major contributors to its increased prevalence [2,3,4,5,6,7]. 

The spread of resistant micro-organisms increases whenever there is less than optimal behaviour or infrastructure as this then facilities the spread of both sensitive and resistant bacteria. e.g., poor infection control, poor sanitation, crowding of animals or people, inadequate clean drinking water, poor animal husbandry, and poor hygiene. The local and global movement of foods, animals, and people is an obvious way resistant bacteria can spread widely. Spread can occur directly from person to person or from one sector to another, e.g., from the agriculture sector to the human sector, and vice versa.

In waterways in India (and elsewhere), multiple different strains of Gram-negative bacteria are often present [3,10]. Transfer of resistance genes carried by these bacteria seems to occur frequently in these waterways.

We believe globally, wherever there is a high prevalence of AMR, this is most likely attributable to “contagion” and this is most likely via poor sanitation and contaminated drinking water. Whenever there is poor administrative governance (e.g., increased corruption levels), this spread is likely to then also occur at much higher rates. Our hypothesis is supported by a strong association with the infrastructure index and the levels of resistant *E. coli* seen because *E. coli* are faecal bacteria and so their presence in water indicates human or animal faecal contamination [15]. Some find this a surprise, but we know that poor water quality and sanitation are linked with many poor health outcomes, so why should we be surprised if poor sanitation levels are also associated with much higher levels of AMR. [14,28,29]. Other examples of how poor infrastructure might facilitate the spread of infections and resistant bacteria is a lack of reliable electricity and refrigeration. If developing countries adopt technology from developed countries, but because of infrastructure, governance, and/or other issues, these frequently may not function adequately. We are aware of a recent example where bed pan washer/sanitizers were no longer functioning. Nurses had to then clean bed pans by hand but without gloves because of poor supply lines. It is easy to see in these situations resistant bacteria can easily spread and that this spread is then a much more important factor in AMR rates seen in comparison to antibiotic usage volumes.

## 6. What is Missing from Studies of Antimicrobial Resistance?

There are major difficulties in nearly all studies on AMR because most make no attempt to capture the importance the One Health, social issues, infrastructure, governance, or even whether the health systems are functioning when trying to look at causal relevance.

Socioeconomic determinants, such as income and access to education, affect antibiotic consumption. Easy access to antibiotics “over the counter” and without prescription is available in many jurisdictions. In Brazil it was shown that socioeconomic determinants are important when restricting over the counter antibiotic sales [30]. Factors that were associated with increased antibiotic consumption were extent of urbanization, higher human development index, density of private health establishments, life expectancy, and percentage of females, lower illiteracy levels and lower percentage of population between five and 15 years of age [30]. 

The multifaceted nature of AMR means we need to understand much better the relative importance of factors other than antibiotic consumption as major drivers for the evolution of resistance. Debased socioeconomic conditions are major enablers for the spread of resistant bacteria and will more easily allow “epidemic-like contagion” of both sensitive and resistant bacteria [5,6,7,15].

The quality of administrative governance, public spending on health, poverty (as measured by gross domestic product [GDP] per person), education, and community infrastructure are known to affect health outcomes [28,29] and likely go a long way to explaining the patterns observed in Figure 1, Figure 2, and Table 1.

Some social enablers of the spread of AMR are difficult to define concepts that challenge conventional modes of data analysis; however, their relevance is transparently obvious. An important example is corruption of a society’s administrative governance. If corruption is poorly controlled, it is much more likely that there will be antibiotic use that is undocumented and so antibiotic volumes used, and their effects, will not be reflected in official figures. With increased corruption there will also be poorer controls on the many ways that resistant bacteria can spread into the environment and/or between animals and people. Inspection, measurement and enforcement of regulations regarding waterways and drinking water will also be compromised. Health regulations for the preparation, transportation, packaging, and storage of food will not be enforced in communities that are administered corruptly. 

Clearly, there are many factors that must be addressed if one wants to control AMR other than just focusing on usage volumes, the types of antibiotic used and usage patterns. There is a desperate need to better understand and quantify the magnitude of, and how social, economic and policy environments enable the spread of AMR. We need to put in place appropriate policy priorities to address these issues. 

## 7. Cultural Determinates are Important

In Europe, socioeconomic determinants, such as population income, demographic structure, density of general practitioners, and their remuneration method, have all been shown to impact outpatient antibiotic use. Richer countries in Europe use more outpatient antibiotics compared to countries with a lower ability to pay. This effect of income has also been shown to be a factor in developing countries. Additionally, supply-side factors and incentives attached to payment schemes (e.g., for physicians in Europe) are likely to be important and need to be considered in government interventions [31]. 

Antibiotics usage and corruption levels in regions of Europe have been studied at a sub-national level by Rönnerstrand and Lapuente [32]. They looked at two measures of corruption (prevalence of corruption in the health sector and prevalence of bribes in the society) and they compared these to the consumption of antibiotics in these different regions. They found a strong positive association of antibiotic use with both of their corruption measures. These were seen both in the health sector and in the community. They concluded that corruption accounts for some of the notable between-region variations in antibiotic consumption seen in Europe.

A previous study looked at the importance of governmental, social, and economic factors as drivers of AMR compared to what are usually considered the main driving factors—antibiotic usage and levels of economic development [19]. Only 28% of the total variation in antibiotic resistance rates between countries could be accounted for by variations in different antibiotic usage volumes in these countries. When the “control of corruption” was included as an additional parameter, then 63% of the total variation in AMR was then explained by the regression. There are indications that “corruption” (i.e., poor behaviour), even in high income countries such as those in Europe, accounts for as much of the variation in AMR rates seen between countries and was as large a factor as antibiotic usage volumes. It is interesting that the monetary measure of poverty (the income level of a country) appeared to have no effect on resistance rates in the multivariate analysis. The analysis suggested that for every improvement in the corruption indicator, this would be associated with a proportionate reduction in AMR. It was also noted that if the private health to public health expenditure ratio was decreased then there was also a likely associated decrease in AMR. This study gives support to the hypothesis that poor governance (e.g., higher corruption levels) contributes to higher levels of AMR and that this correlates much better with the observed AMR rates than antibiotic usage volumes. The study concluded that addressing social issues, such as corruption and improving administrative governance, would lead to a reduction in AMR [19].

In our recent study we found that globally poor administrative governance and corruption are strongly correlated with high levels of AMR [15], a result also seen in a previous study in Europe [19]. The previously discussed effects of corruption on AMR levels are, however, likely to be indirect.

## 8. Conclusions

AMR continues to rise globally and is associated with increased deaths as well as increased suffering for people. However, there are many things we can do to improve the situation. These are best captured within “One Health”, although it will be important that the “One Health” concept does not predominately focus on antibiotic use and levels in the human, agriculture, and environmental sectors, but adequately addresses “contagion” and includes social and economic determinants. 

Global evidence shows a pattern where poor infrastructure and corrupt administrative governance are associated with lower levels of antibiotic consumption yet higher prevalence of AMR. Even in high income countries, where the focus has been mainly on antibiotic consumption as the most important factor contributing to AMR. However, antibiotic usage volumes explain only a small portion of the observed AMR levels. Contagion, rather than antibiotic usage volumes, is the major factor contributing to the variations in antibiotic resistant levels across countries. Contagion is, in turn, affected by many social determinates and economic factors. 

This new insight has major policy implications. Interventions focusing on decreasing antibiotic consumption alone are not likely to be sufficient, especially in LMICs, because contagion is likely the dominant factor affecting the prevalence of higher levels of AMR.

We need to improve all the economic and social factors that contribute to “contagion” and so then make spread less likely to occur. This needs to include better sanitation, better access to clean water, better housing, less crowding, safer foods, less transmission in hospitals by adopting better infection control and prevention practices, as well as improving administrative governance (i.e., lower corruption). We also need an increase in public expenditure on health if we are to better tackle AMR on a global scale. Better approaches that take into account cultural contexts are required to better facilitate successfully approaches to what and how antibiotics are used in people and restrictions on use that are needed. It is also important that these approaches extend to management of food animals, especially for critically important antibiotics. 

Future research and policy formulation must much better recognise and address the importance of cultural, social, and economic determinates that drive contagion, because it is the complex amalgam of these factors that enable and facilitate the spread of AMR. 

## Figures and Tables

**Figure 1 antibiotics-08-00086-f001:**
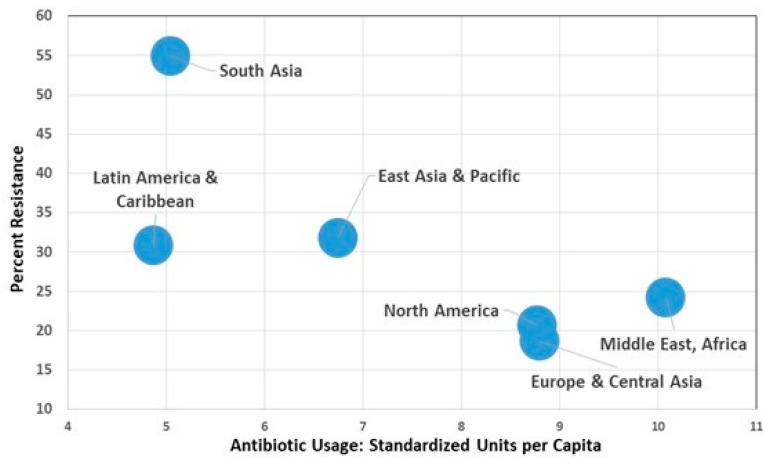
Percent resistance in *E. coli* to third-generation cephalosporins (3GCeph) and fluoroquinolones (FQ). Source Data: Collignon et al. (2018) [15] and figure from Collignon and McEwen [6].

**Figure 2 antibiotics-08-00086-f002:**
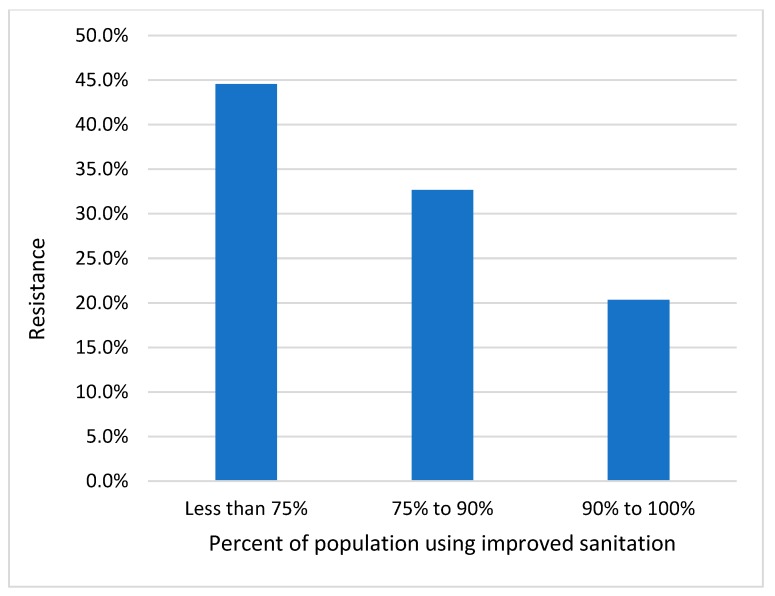
Improved sanitation versus resistance percentage of *E. coli* to 3rd-generation cephalosporins and fluoroquinolones. Globally, these antibiotic resistance levels are inversely related to usage volumes but strongly related to sanitation levels. Source Data: Collignon et al. (2018) [15] and World Health Statistics, WHO, (2011) [16].

**Table 1 antibiotics-08-00086-t001:** Income effects: antibiotic usage and antimicrobial resistance levels.

World Bank Income Group	Antibiotic Usage CDDEP Standardized Units	Percent Resistance *E. coli* to 3GCeph&FQ
High income countries	8.5	18.3
Upper middle-income countries	7.2	31.1
Lower middle-income countries	6.9	42.6
Grand Total	7.9	25.6

Source: Percent resistance *E. coli* to third-generation cephalosporins (3GCeph) and fluoroquinolones (FQ). CDDEP is the Centre for Disease Dynamics, Economics and Policy. Source Data: Collignon et al. (2018) [15].

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
