# Peer review of "Socioeconomic Enablers for Contagion: Factors Impelling the Antimicrobial Resistance Epidemic"

_antibiotics, 2019, doi:10.3390/antibiotics8030086_

Round 1

Reviewer 1 Report

Thank you for your revision and comments. You are right, I missed that this was an invited commentary. That along with having a results and discussion section made it even more challenging. 

I am content with the revision  

Author Response

Yes, this paper was an invited review/commentary rather than a primary research paper and it is now more appropriately presented in that format. We would like to thank the reviewer for re-looking at our paper and approving the content and format of the revised manuscript.

Reviewer 2 Report

Line 14 -  Omit the word ‘only’ in this sentence?  The word feels superfluous and slightly contradictory given that the paper goes on to unpick the complexity of AMR and its drivers.   

Lines 45-54:  I’m uncertain of the benefit of using the term ‘superbug’ in this section.  ‘Superbug’, as a term, contributes to an (often media-fuelled) narrative on AMR, which focus on catastrophe which can be problematic when trying to bring about the political, social and economic changes required (See Nerlich and James, 2008).  Using the term interspersed with resistant bacterial pathogens is slightly confusing.   The authors may wish to acknowledge the roots of the term ‘superbug’ or stick to resistant bacterial pathogens as the descriptor. 

Line 56:  As above in the abstract (line 14), and given the extent of science currently happening in relation to factors that drive AMR, I would suggest omitting the word ‘only’ here, to write ‘there appears to be two main factors …’  

Line 65: It might be useful to distinguish here between ‘over-prescribing’ and ‘over-use’ and in relation to human and animal health in the examples given here.

Line 77-78: Fascinating finding.

Line 66: ‘Contagion’ – the authors could usefully describe what is meant by the term ‘contagion’ in this context.   Particularly, as it goes on to be a key thread within the paper.

Line 138 AMC needs to be written out in full before being abbreviated.

Lines 124-164: the point which is being made is that there has been too much focus on antibiotic use and not enough on the conditions of contagion.  The last two paragraphs, which focus again on use rather than contagion, perhaps detract from the interesting point being made in this section.  Unless, that is that, the point being made is about ‘undocumented use’ and that this could be driving resistance and affecting the data on the relationships between use and resistance (which presumably can only be based on documented antibiotic use)? 

P209:  As above, could the ‘antibiotic use that is undocumented’ be an important factor shaping the data on the relationships between use and resistance?

P259: Would be useful to describe what is meant by ‘One Health’ within this context.

P278 The focus in this final sentence on ‘poor behaviour’ feels like an anomaly from the main argument of the paper that the drivers relate to ‘contagion’ and are at the level of infrastructure (in particular around political systems/corruption) – unless the ‘poor behaviour’ relates to the processes (or lack) of governance enacted by the officials involved in corruption?

Author Response

We would like to thank this reviewer for his/her very extensive efforts; not only in their thorough reading of the revised manuscript but the obviously large amount of time spent on it and with so many helpful and thoughtful suggestions for improvements.

Comments and Suggestions for Authors

Line 14 -  Omit the word ‘only’ in this sentence?  The word feels superfluous and slightly contradictory given that the paper goes on to unpick the complexity of AMR and its drivers.   

We agree and we have modified the sentence.

Lines 45-54:  I’m uncertain of the benefit of using the term ‘superbug’ in this section.  ‘Superbug’, as a term, contributes to an (often media-fuelled) narrative on AMR, which focus on catastrophe which can be problematic when trying to bring about the political, social and economic changes required (See Nerlich and James, 2008).  Using the term interspersed with resistant bacterial pathogens is slightly confusing.   The authors may wish to acknowledge the roots of the term ‘superbug’ or stick to resistant bacterial pathogens as the descriptor. 

We agree that it might be better to avoid the term Superbug in this scientific paper and we have removed and/or modified the term as requested throughout the document.

Line 56:  As above in the abstract (line 14), and given the extent of science currently happening in relation to factors that drive AMR, I would suggest omitting the word ‘only’ here, to write ‘there appears to be two main factors …’  

We agree and we have modified the sentence.

Line 65: It might be useful to distinguish here between ‘over-prescribing’ and ‘over-use’ and in relation to human and animal health in the examples given here.

Agree this might be useful and have added text to better explain this. But believe it better if added at line 159 to 163.

Line 77-78: Fascinating finding.

Thank you

Line 66: ‘Contagion’ – the authors could usefully describe what is meant by the term ‘contagion’ in this context.   Particularly, as it goes on to be a key thread within the paper.

Agree, and we have better defined the term “contagion” early in the paper now see line 66. We have also added it in effect to the abstract see line 13/14

Line 138 AMC needs to be written out in full before being abbreviated.

Agree and we have modified the sentence where the term is first used. See line 169.

Lines 124-164: the point which is being made is that there has been too much focus on antibiotic use and not enough on the conditions of contagion.  The last two paragraphs, which focus again on use rather than contagion, perhaps detract from the interesting point being made in this section.  Unless, that is that, the point being made is about ‘undocumented use’ and that this could be driving resistance and affecting the data on the relationships between use and resistance (which presumably can only be based on documented antibiotic use)? 

Agree and very valid point. We have moved a paragraph and made modifications to better reflect the point the reviewer is making.

P209:  As above, could the ‘antibiotic use that is undocumented’ be an important factor shaping the data on the relationships between use and resistance?

Agree and very valid point. We have modified some of the text to better reflect the important point the reviewer is making. See line 243.

P259: Would be useful to describe what is meant by ‘One Health’ within this context.

Agree. We have added text to better define and explain this early in the text - see lines 32-34.

P278 The focus in this final sentence on ‘poor behaviour’ feels like an anomaly from the main argument of the paper that the drivers relate to ‘contagion’ and are at the level of infrastructure (in particular around political systems/corruption) – unless the ‘poor behaviour’ relates to the processes (or lack) of governance enacted by the officials involved in corruption?

Agree and very valid point. We have modified the sentence to better reflect the point the reviewer is making re the importance of contagion. See lines 301-303.

Reviewer 3 Report

This current submission is not adding any significant novel ideas to the field of antibiotic resistance. Much of the included work can be directly found in multiple other manuscripts published by P. Collignon.

Author Response

Reviewer 3

Comments and Suggestions for Authors

This current submission is not adding any significant novel ideas to the field of antibiotic resistance. Much of the included work can be directly found in multiple other manuscripts published by P. Collignon.

We think the reviewer has misunderstood the purpose of this paper. This paper is an invited  commentary rather than a research paper presenting completely new and novel ideas.

There are however many novel aspects in our paper. Data has been analyzed and presented in ways not previously published. All the figures and tables are “new” in that the data used for these (ref 15) has not been analyzed in this way previously. We have now added a new section on crowding as well after analyzing and presenting some more global data.

This Commentary is also novel because it addresses frequently-neglected public policy relevant aspects of AMR contagion. It is written in a manner to be readily understood by medically trained readers, even those without expert statistical knowledge. The paper is not intended to be only about technical analysis.

We would like to thank this reviewer for having spent so much of his/her time in keeping up and reading of the very many papers that Peter Collignon has written on the issue of AMR, over many decades. Those past papers have involved all aspects of AMR but have mainly focused on antibiotic use/overuse in people and especially overuse/abuse in food animals as well as the effects of water and the environment in spread and on “One Health” aspects. The other major focus has been on healthcare associated infections and more recently corruption. Many of these are referenced in this current paper, to show where previous ideas have come from.

As commented earlier, even thought this is not a primary research paper, we disagree with the reviewer that this is not adding any “significant novel ideas to the field of antibiotic resistance”. This paper was an invited review, with a request to focus on social issues to do with AMR, as obviously the editors felt this has not been covered very well in previously published literature - not only in paper Peter Collignon has co-authored with others, but in the literature in general. We agreed with the editors’ viewpoint and hence why we agreed to write this current paper. We provided an outline and drafts of the paper to the editors before submitting it to ensure it followed what was asked of us.  This request for this current paper was made to us by the editors followed the publication of our recent Lancet Planetary Health paper, where we made the point that corruption and behavior were also likely an important part of the issue (i.e. social issues) as well as the importance of infrastructure. However, in that Lancet paper (as in others that Peter Collignon has written), social, economic and behavior issues are just touched on, rather than getting any extensive coverage and being the main focus of any of the papers themselves (e.g. the Lancet paper concentrates predominantly on spread and contagion). Hence, we believe this paper is a unique and novel contribution to the literature and additionally we have followed the content and structure as requested for us to follow by the editors.

This while we greatly appreciate the reviewer’s reading, interest and following all of Peter Collignon’s  previously published publications, we don’t agree that this current paper is not making a very useful new contribution. Our presumption is with time this current paper will be extensively used and referenced by others - and which will be the ultimate test of the likely significant novel ideas to the field of antibiotic resistance. We believe this paper is a significant new contribution to the literature on AMR.

Reviewer 4 Report

The study by Peter Collignon and John J. Beggs, shows data about antimicrobial resistance, a growing global problem. They have described two factors that drive antimicrobial resistance: the volumes of antimicrobials used and the spread of resistant micro-organisms. They also claim that poor infrastructure and corruption levels are highly and positively correlated with antimicrobial resistance levels. They conclude that antimicrobial resistance could be better controlled.

I think that the review is not fully innovative and I find it partly obvious since the literature is already rich in this type of data. For this reason, in its current form, this work is not yet ready for publication.

Major comment

1. It is opinion of the referee that the article must be improved and enriched with other appealing and innovative data… This could make the reading of this review more likely. I found it a bit obvious and repetitive.

Minor points

1. E. coli is always written without a space between E. and coli (see lines 40, 41, 48, 51, 182, 183 …..)­;

2. At the end of line 44 the point is missing;

3. On line 47, E. coli should be written by using cursive character;

4. In the Figure 1 and 2, there are a missing space between E. and coli;

5. In the text, Table is sometimes indicate as a capital and sometimes as lowercase letter;

6. On the line 120, there is an extra comma;

7. From line 133 to line 139, the paragraph layout is not justified.

Author Response

Reviewer 4

Comments and Suggestions for Authors

The study by Peter Collignon and John J. Beggs, shows data about antimicrobial resistance, a growing global problem. They have described two factors that drive antimicrobial resistance: the volumes of antimicrobials used and the spread of resistant micro-organisms. They also claim that poor infrastructure and corruption levels are highly and positively correlated with antimicrobial resistance levels. They conclude that antimicrobial resistance could be better controlled.

I think that the review is not fully innovative and I find it partly obvious since the literature is already rich in this type of data. For this reason, in its current form, this work is not yet ready for publication.

As commented in our repose to reviewer 3, we disagree that this review is not innovative and that the literature is already rich in this type of data. Our paper followed  a request by the editors for a commentary type paper, and followed the publication of our recent Lancet Planetary Health paper, where we made the point that corruption and behavior were also likely an important part of the AMR issue. However, in that Lancet paper, these social issues are just touched on rather than getting any extensive discussion nor being the main focus of the paper itself (e.g. the Lancet paper concentrates predominantly on spread and contagion - ref 15). This paper was an invited commentary, with a request to focus on social issues to do with AMR - as obviously the editors felt this has not been covered very well previously in published literature. We agreed with the editors’ viewpoint and hence why we agreed to write this current paper and where we provided an outline and drafts to the editors before submitting it. 

There is also many novel aspects to this paper as all the figures and tables are “new” in that the data used for these (ref 15) has not been analyzed in this way previously.

This Commentary addresses frequently-neglected public policy relevant aspects of AMR contagion. It is written in a manner to be readily understood by medically trained readers, even those without expert statistical knowledge. The paper is not intended to be about technical analysis. Thus, we disagree that this is already well covered in the literature already or that this paper is not novel.

Major comment

1.      It is opinion of the referee that the article must be improved and enriched with other appealing and innovative data… This could make the reading of this review more likely. I found it a bit obvious and repetitive.

We disagree. This is was an invited review not a primary research paper. So it is not the format where “other appealing and innovative data” can normally be introduced. New data needs generally to be in a primary research paper rather than an invited review. This was also not as far as we have understood, the request nor intent of the editors when our paper was invited for submission.

However we have put many novel aspects into this paper re data analysis. All the figures and tables, including looking at crowding are “novel” and “new”, because the very large data set used for the Lancet paper (ref 15) has not been analyzed in this way previously, and is showing that many social - economic issues, such as crowding (see our figures and tables)  may be much more important than currently appreciated as factors in AMR.

Minor points

1.      E. coli is always written without a space between E. and coli (see lines 40, 41, 48, 51, 182, 183 …..)­;

We agree, and believe this is now fixed throughout the paper.

2. At the end of line 44 the point is missing;

Agree, now fixed.

3. On line 47, E. coli should be written by using cursive character;

Agree, now fixed.

4. In the Figure 1 and 2, there are a missing space between E. and coli;

Agree, now fixed and E.coli should be in italics.

5. In the text, Table is sometimes indicate as a capital and sometimes as lowercase letter;
Agree. We presume always with a non-capital T but will leave that to editorial staff re the preference of the journal.

6. On the line 120, there is an extra comma;

Agree, now fixed.

7. From line 133 to line 139, the paragraph layout is not justified.

Agree, now fixed.

Round 2

Reviewer 3 Report

Paragraph indentation is inconsistent and paragraph structure should be adjusted such that single sentences are not standing alone. 

Line 42 - should say pan-resistant, delete else and

Line 44 missing period

Line 65 - delete via

Line 113 - should be of not on

Line 141 - should say consistent not consonant 

Line 141 - comma after level

Author Response

Reviewer 3

Comments and Suggestions for Authors

Paragraph indentation is inconsistent and paragraph structure should be adjusted such that single sentences are not standing alone.

Thank you. We will however defer to editorial staff on any changes needed re indentation etc, to fit in with the format that the Journal requires.

Line 42 - should say pan-resistant, delete else and

Line 44 missing period

Line 65 - delete via

Line 113 - should be of not on

Line 141 - should say consistent not consonant

Line 141 - comma after level

Thank you for finding all these corrections needed to improve punctuation and spelling etc. We appreciate the effort you made to read the paper so thoroughly to identify these. We have made all the changes suggested.

Reviewer 4 Report

The answers of the authors are satisfactory for me.

Author Response

Reviewer 4

Comments and Suggestions for Authors

The answers of the authors are satisfactory for me.

Thank you for your time in reviewing our paper and approving this final version.

This manuscript is a resubmission of an earlier submission. The following is a list of the peer review reports and author responses from that submission.

Round 1

Reviewer 1 Report

Line 31: state some examples of why AMR much worse in LMICs

58: pls provide additional references to back this statement as ref 7 is a clinical perspective written by the authors 

123-126: this is not sufficiently backed up. It can be argued that actually there is significant emphasis on infection prevention and control 

No methods. 

I am unclear how there are results without methods, even if of approach to review. The results presented are from previously published article. 

Whilst I agree that socio and economic factors are important,  I am unclear how this article differs sufficiently from ref 15, also led by authors 

Reviewer 2 Report

A very superficial review, and lacking the right depth necessary for such a subject. No antibiotic resistance models are reported to support the proposed arogmentations.